# Sustainability of Commercial Banks Supported by Business Intelligence System

**Remigiusz Tunowski**

Faculty of Finance and Management in Gdansk, WSB University in Gdansk, 80-266 Gdańsk, Poland; rtunowski@wsb.gda.pl

**Abstract:** This article was focused on establishing whether Business Intelligence (BI) systems provide sustainability to commercial banks by influencing their financial condition. As part of the search for a solution to the research problem, a hypothesis was formulated which assumes that the use of the Business Intelligence management system improves the financial condition of commercial banks. To assess this impact, a novel comparative method was used, which assumed comparing financial condition indicators in three aspects: before and after the implementation of the Business Intelligence system (comparison over time), with average indicators of a group of banks (comparison to the industry), with reference to changes in the overall economic situation. As a result of the method used, a synthetic indicator of the impact of using Business Intelligence (ABI) was calculated. This study was conducted in relation to six out of the thirteen largest commercial banks listed on the Warsaw Stock Exchange in 2020, which have implemented the Business Intelligence system since 2001. The assets of the examined banks cover 60% of the assets of commercial banks in Poland. As a result of the study, a positive impact of using the BI system on selected areas of the financial condition of commercial banks was identified. In particular, this impact relates to areas of productivity, the quality of assets and liabilities, profitability and debt. The generalized results of this study allow for the determination of cause and effect relationships between the use of the BI system in commercial banks and the improvement of the financial condition indicators as well as sustainability banking.

**Keywords:** Business Intelligence; Business Intelligence impact; bank; financial condition; sustainability banking

---

## 1. Introduction

The growing needs of the organization in the field of analysis, the interpretation and processing of data have led to a need to build information systems integrating information from dispersed sources into one homogeneous and transparent information portal. The Business Intelligence (BI) class system has proved to be helpful in achieving this objective, and has become an inseparable element of doing business in the 21st century [1]. The definition of a Business Intelligence system has been evolving over the years. In 1958, H.P. Luhn, working for the IBM Corporation, defined the term Business Intelligence for the first time as "*The ability to understand the relationship between the facts presented in such a way as to take action towards the set goal*" [2]. Currently, it refers to a broad concept of business analytics. Business intelligence class systems are assumed to have a user-oriented process of collecting, exploring, interpreting and analyzing data that leads to streamlining and rationalizing the decision-making process. These systems support the managerial staff in taking business decisions, whose main goal, in turn, is a sustainable increase in the company value [3].

This publication attempts to identify the impact of using Business Intelligence systems on selected indicators of the financial condition of commercial banks in Poland. The financial condition of an organization is very important from the perspective of its operation and in the view of the goals set by its

stakeholders. In the literature, the term *financial condition* is identified as the financial situation, which is the result of economic decisions taken by the company and the opportunities for its development related thereto. The International Accounting Standards (IAS) define *financial condition* as the ability to generate income or raise funds from other sources, including funds acquired during its current operations. The financial condition is perceived through the prism of its five determinants, namely liquidity, productivity, financial support, profitability and the position in the financial market. In the literature, the study of financial condition is also treated as a synonym of financial analysis. The financial condition is then seen as the result of the company's financial management, and thus it is one of the determinants of the entity's management processes. There is therefore feedback between financial health and financial management in the enterprise. This condition is influenced by many factors that can be divided into the external ones, i.e., economic situation, cost of capital, interest rates and the internal ones, i.e., achieved revenues and the incurred costs, the quality of management, including systems supporting information management process, e.g., the Business Intelligence class [4,5].

In the literature, the subject of the impact of Business Intelligence systems on organizations has been the subject of research of many authors. However, the problem of measuring the impact of BI is a challenge, because we do not always see at first glance the benefits of implementing and the subsequent use of the Business Intelligence system. To date, the subject of the impact of using Business Intelligence systems on organizations has been recognized in two ways [6]:

- quantitative analyses—examining the impact of using the Business Intelligence (BI) system on indicators that illustrate the economic situation of an organization [7–18];
- case studies—studying the impact of the BI system on the functioning of individual organizations [19–30].

An interesting example of one the most recent studies is the empirical analysis based on the structural equation modeled with data collected from 88 Italian small and medium enterprises (SMEs). The authors tested whether the analytical capabilities had a positive impact on the firms' performances. The findings show that the firms that developed more big data capabilities than others, both technological and managerial, increased their performances [17].

In 2018, a study was conducted in Brazil, to identify the contemporary status of big data analytics, occurring at various management levels of organizations and supply chains in domestic firms. The results of this study demonstrated the complications and hindrances in the big data analytics adoption and pointed out the relationship between the big data analysis knowledge and the supply chain levels [18].

Most research regarding the influence of the BI system on the organizations' condition has shown its positive impact, hence an analogy concerning banks can therefore be assumed. The analysis of the literature also shows the year-to-year growth of the use of the BI system, increasingly applied in new industries, and the industries already familiar with the system applying it in new ways. In turn, the case studies analyzing the implementation of BI systems indicate that the results of the systems' application are hard to measure and bring fruition in the longer perspective.

Although organizations and industries willingly integrated the Business Intelligence system at a significant scale, more than 70% of Business Intelligence projects failed to bring the expected effects [31–33]. Due to the above, the understanding of how the Business Intelligence system affects the financial conditions of an organization must be accompanied by the identification of the determinants, which would significantly affect its adoption. There are studies, described by Ahmad, related to the determinants connected with Business Intelligence system adoption. A structured literature review, based on 84 selected articles published between 2011 and 2020, have utilized a wide range of frameworks, models and theories to investigate the major determinants for the Business Intelligence system adoption. In the literature, there are three main theories used by studies regarding the Business Intelligence system adoption: 48.50% of studies have used the diffusion of innovation (DOI), 35.40% have applied a technology–organization–environment (TOE) framework while institutional theory has been described

by 32.25% of studies [34]. The diffusion of innovation theory lists five main determinants influencing the rate at which any innovations are adopted: relative advantage, compatibility, complexity, trialability, and observability. All of those determinants, apart from complexity, are mostly positively related with the adoption rate [35]. Technology–organization–environment is a framework also frequently applied in the adoption of Business Intelligence system. The theory describes the technological, organizational and environmental dimensions that organizations take into consideration when implementing new technologies [36]. The technological dimension comprises internal and external technologies that may apply both the tools and processes essential to the organization. The organizational dimension relates to the size of the company, the scope and centralization levels, the number of available resources (e.g., staff and other resources), and the managerial structure of the organization. Lastly, the environmental dimension refers to the industry structure, the macroeconomic context, size, the competition and the relevant governmental policies/regulations [37,38]. According to the institutional theory, it is the pressures from customers, suppliers, trading partners, competitors, and governmental bodies that can affect a company's decision to implement a Business Intelligence system [39].

Ahmad also presented an interesting issue concerning the sector-related distribution of the analyzed papers describing the adoption and implementation of Business Intelligence. Authors found that they mostly concentrated on the banking sector (23%), multiple companies (18%), small and medium enterprises (SMEs) (17%), the telecommunication sector (9%), and the healthcare sector (7%). Sectors that had attracted less attention of the researchers included insurance companies (5%), retail chains (3%), supply chains (3%), logistic services (3%), BI vendor companies (3%), electronic industry (4%), and academia (3%) [34].

However, even if there are studies related to the adoption of Business Intelligence in the banking sector [40,41], there is still a lack of studies on the impact of the use of Business Intelligence. Especially, there is a lack of studies related to the impact of the BI system on the financial conditions of the commercial banks. The relevant research literature is dominated by enterprise-related case studies, and financial institutions are rarely examined. This omission applies in particular to banks that form one of the key sectors of the modern economy [42,43]. Apart from the study published by the author of this article in 2019 [44], in Poland there have been no long-term studies assessing the impact of BI systems on organizations, in particular on commercial banks. Studies on the impact of BI on enterprises in Poland have mainly included case studies only [45–47]. There has been no research on the impact of these systems on the financial condition of organizations. As indicated by previous studies, there has been a visible shortage of research regarding the impact of the implementation and the use of BI in commercial banks. In particular, it concerns the financial condition of banks operating in the Polish economic environment.

This study hopes to enrich the recent literature because it focuses on the rarely studied long-term impact of Business Intelligence systems in the banking sector. However, it also intends to improve the understanding of practitioners' decision-making processes to leverage maximum value from the adoption of Business Intelligence system. It is also worth noting that to date, no uniform method of assessing the impact of BI on the condition of an organization has been identified as a result of known research. The authors use very diverse methods, both qualitative and quantitative. This article describes a novel comparative method of analyzing changes in the financial condition of the bank following the implementation of the BI system, based on comparisons of time- and space- related financial indicators.

This publication was an attempt to recognize the impact of the implementation of the Business Intelligence system on the financial condition indicators in large Polish commercial banks listed on the Warsaw Stock Exchange in 2020. Finally, six banks that have implemented the BI system were analyzed.

The study included Polish commercial banks that met the following three criteria of qualification:

- operational in the year of the research (2020);
- present on the Warsaw Stock Exchange;

- provided public information on the date of implementation of the Business Intelligence class system after 2001 and until 2018.

Finally, the implementation of the Business Intelligence class system in six large Polish commercial banks, out of the thirteen listed on the Warsaw Stock Exchange in 2020, was analyzed. The individual banks were successively named with letters from A to F. The assets of the selected banks covered about 60% of all assets of commercial banks in Poland, therefore, the results of the survey can be transposed to the commercial banking sector in Poland in general. The financial data used in this study covered periods both before and after the implementation of the Business Intelligence system. The implementation of the BI system was the moment of launching the Business Intelligence class system that for the purpose of this publication was the date of the bank informing the public of the implementation of a such system. However, the use of the BI system is a process during which a given organization (bank) uses the functionalities of the BI system. The periods of implementing the Business Intelligence system in the particular banks are presented in Table 1, below.

**Table 1.** Quarter of the implementation of the Business Intelligence system in each bank.

| Bank | Quarter of Business Intelligence System Implementation |
| --- | --- |
| Bank A | 3 Q 2011 |
| Bank B | 2 Q 2011 |
| Bank C | 1 Q 2007 |
| Bank D | 2 Q 2006 |
| Bank E | 2 Q 2013 |
| Bank F | 1 Q 2014 |

Source: own elaboration.

The implementation of the Business Intelligence system in the six banks examined took place within eight years. Two banks decided on the BI system in 2006–2007, i.e., shortly before the apogee of the global financial crisis, associated with the fall of Lehman Brothers, an American investment bank, in September 2008. Two more banks decided to implement it in 2011, while the last implementations of the Business Intelligence system took place in 2013 and 2014.

The research concerning the application of the BI system in organizations indicates the development of analytical skills, which is the beginning of changes resulting in the improvement of the operation of those organizations. Banks represent a particular type of organization; they collect and process huge volumes of data that can be analyzed (with the information contained there) by the BI system much more efficiently, and as a result, can be used in taking better management decisions.

The research results presented above are a premise to hypothesize that the implementation of Business Intelligence systems has improved the financial standing of the commercial banks listed on the Warsaw Stock Exchange in 2020 and in a wider perspective, could enhance the sustainability for the banks using BI. It is consistent with the results of the 20 scientific studies that conclude the positive interconnection between environmental, social and governance (ESG) factors and the financial performance [48]. Moreover, a Deutsche Bank report proves that there are a number of reasons why financial, environmental and social objectives can be consistent with each other and consideration for ESG criteria can increase shareholder value [49].

However, the findings from the study made on the Polish banking sector for the period 2008–2015 are not consistent. The authors analyzed the connection between corporate social responsibility and financial performance. The result identified the positive relationship between human resources and financial performance, but the negative relationship was identified between the community involvement, product, customers and the financial performance [50]. A study based on the data collected from 166 banks from 31 countries shows that financial stakeholders have exerted great pressure to adapt bank management systems and take environmental aspects into account [51].

The interesting study about the sustainability of banking business models was performed on sixteen European financial institutions in 2019. The authors revealed important determinants of performance sustainability, which were value proposition, core competencies, financial aspects, business processes, target customers, resources, technology, customer interface and partner network. The results showed that the sustainability of business models of Norwegian and German banks was higher compared to other countries (including Poland). The Polish banking sector was ranked third in the sustainability of the business model [52].

Sustainable banking has become an important topic in recent years. One of the reasons is that banks had their reputation undermined as a result of financial crisis in 2008. The top management realized that taking actions connected with sustainable banking could restore the position of those financial institutions. This is one of the reasons why sustainable banking has become an important topic to managers as well as researchers. A recent systematic review, based on 255 publications, connected with the topic of sustainability banking, showed that the number of papers has increased dramatically in the last ten years. In the period 2015–2019, the number of studies doubled compared to 2009–2014 [53].

This publication is a continuation of the research described by the author in his book published in 2019 and related to the impact of Business Intelligence systems on the financial condition of commercial banks in Poland based on dynamic panel data models [44].

## 2. Materials and Methods

The quarterly financial statements of the commercial banks listed on the Warsaw Stock Exchange in 2020, obtained from the Notoria database (the Notoria database contains an updated, unified format of financial statements for all companies listed on the Warsaw Stock Exchange, available online at https://ir.notoria.pl/oferta_mssf.html) were the source of data used for calculations. Since the variety of financial indicators, when excessive, may lead to an obscure picture of the financial situation of the bank, they need to be selected. For the purpose of this study, the selection of indicators was made on the basis of financial analysis-related literature, according to the two criteria specified below.

The indicators commonly described in the literature related to the financial condition ratio analysis were selected first. Those described by many authors, therefore being a common part, were chosen, constituting a group of indicators widely known and described as significant from the point of view of the financial condition. As a result, the indicators analyzed not only in the banking sector, but also in other industries, were selected [54–57].

Banking sector specific indicators were selected as the second. Attention was focused on indicators providing information on the relationships between such major financial categories regarding banks as deposits, loans, operations with the National Bank of Poland and related to the result of banking activity.

Table 2 presents the affiliation of indicators to individual areas of the financial condition assessment and their mathematical formulas.

The indicators of the banks' financial condition finally selected for the study consisted of six areas of assessment: liquidity (2 indicators), quality of assets and liabilities (3 indicators), debt (2 indicators), productivity (3 indicators), profitability (3 indicators), and capital adequacy (1 indicator). All calculations contained herein were based on the specified set of the aforementioned 14 indicators of the banks' financial condition.

For a detailed assessment of the impact of the implementation and use of BI on the financial condition of a specific bank, a three-stage comparative method based on calculated financial ratios was developed. The comparative method assumes the construction of a synthetic indicator for assessing the quality of implementation and application of the Business Intelligence system (ABI). A similar method was used by the author in previous research, however, it was limited to a short specific period and to one bank for the chosen period [44].

**Table 2.** Indicators of the bank's financial condition used in the study.

| Area | Indicator | Formula |
|---|---|---|
| Capital adequacy | Solvency ratio | $\frac{Equity\ capital}{Risk\ (weighted\ assets)}$ |
| Quality of assets and liabilities | Non-profit asset financing ratio | $\frac{Equity\ capital}{Total\ assests} \times 100\ [\%]$ |
| | Equity ratio | $\frac{Non-profit\ assets}{Equity\ capital} \times 100\ [\%]$ |
| | The share of loans in assets | $\frac{Receivables\ from\ customers}{Total\ assets} \times 100\ [\%]$ |
| Liquidity | Coverage of receivables from customers | $\frac{Receivables\ from\ customers}{Liabilities\ from\ customers} \times 100\ [\%]$ |
| | Cash payables ratio | $\frac{Cash\ at\ the\ beginning\ of\ the\ period}{Liabilities} \times 100\ [\%]$ |
| Productivity | Operating cost to assets ratio | $\frac{Bank\ and\ general\ management\ expenses}{Result\ on\ banking\ activities} \times 100\ [\%]$ |
| | Operating expense ratio to banking income | $\frac{Bank\ and\ general\ management\ expenses}{Total\ assets} \times 100\ [\%]$ |
| | Asset utilization rate | $\frac{Revenues\ from\ core\ operations}{Total\ assets} \times 100\ [\%]$ |
| Profitability | Operating profit margin | $\frac{Net\ profit\ (loss)}{Equity\ capital} \times 100\ [\%]$ |
| | Return of assets (ROA) | $\frac{Net\ profit\ (loss)}{Total\ assets} \times 100\ [\%]$ |
| | Return on equity (ROE) | $\frac{Operating\ result}{Revenues\ from\ core\ operations} \times 100\ [\%]$ |
| Debt | Ratio of liabilities in equity | $\frac{Liabilities\ to\ customers}{Total\ assets}$ |
| | Customer payables ratio | $\frac{Liabilities}{Equity\ capital}$ |

Source: own elaboration based on [54–57].

The construction of the synthetic indicator can be divided into three stages:

- Stage I—making comparisons of the financial condition indicators, in parallel and in three perspectives (related to time, industry and to the general economic situation);
- Stage II—awarding a component grade to each comparison;
- Stage III—calculating the value of the synthetic assessment indicator for the implementation of the Business Intelligence system.

In the first stage, the comparative method was based on the three parallel comparisons of each of the financial condition indicators:

1. Before and after the implementation of the BI system (comparison over time);
2. With average bank group ratios (industry comparison);
3. With reference to changes in the general economic situation.

The first comparison assumes juxtaposing the indicator values for the first period before and the second period after the implementation and application of the Business Intelligence system. The second period began three months after the BI implementation and lasted until the end of the analyzed period. This comparison indicates whether the indicator improved or deteriorated after the BI implementation.

Two further comparisons will allow to single out the impact of using BI on changes in a given indicator among the factors related to changes in the industry and the entire economy.

Thus, the second comparison concerned the average values of the indicator after the implementation of the BI system contrasted with the synthetic indicators characterizing the situation of a group of banks in a given period. The average values of the indicators after the implementation of BI were calculated as the arithmetic average of the period of the BI system being applied. The ratios for the group of banks were calculated as the weighted average of the ratios in which the assets of

individual banks were weights. This comparison made it possible to refer any improvement in the condition of the audited bank to changes in the situation related to the closest competition—large commercial banks.

The third comparison was intended to examine the dynamics of the indicator during the period of applying the BI system, set against the dynamics of growth of GDP at the same time. The year-to-year dynamics of the indicator for each quarter will be compared with the corresponding dynamics of the GDP. This comparison allows, to some extent, to "isolate" the possible improvement in the condition of the audited bank due to the implementation of BI from changes resulting from the improvement of the overall economic situation.

In the second stage, each of the financial condition indicators was awarded component grades. As in Stage I, these grades were awarded in three parallel takes. During each of the comparisons, a component grade was awarded. Each comparison was calculated using a quantitative method to prevent any judgmental approach. Additionally, each component grade has a specific materiality threshold which directly assigns a comparison to the specific component grade. The grade may take the following forms:

- Positive (P)—favorable situation, a positive change was observed affecting the examined indicator of the bank's financial condition, and it was defined that a "positive" rating would be given when the assessed rate had a minimum value of 3% more favorable than the compared;
- Without changes (WC)—no significant influence on the indicator value was observed, it was defined that "without changes" was a situation in which the assessed indicator had values in the range of +/−3% of the compared values;
- Negative (N)—unfavorable situation, a negative change was observed affecting the examined indicator of the bank's financial condition, it was defined that a "negative" rating would be given when the assessed ratio had a minimum value of 3% less favorable than the compared.

Then, based on component grades, appropriate point grades were awarded, which, depending on the results of the comparisons, were expressed using a seven-point Likert scale [58]. The criteria for awarding individual grades were presented in Table 3 below.

**Table 3.** Grade criteria for the Likert scale.

| Grade | Grade Criterion |
|:---:|:---:|
| 1 | No a positive result of the indicator comparison in 3 out of 3 comparison cases |
| 2 | No positive result of the indicator comparison in 2 of 3 comparison cases and "without changes" for the 3rd comparison |
| 3 | No positive result of comparing the indicator in 1 of 3 comparison cases and "without changes" for the other 2 comparisons, or No positive result of index comparison in 2 out of 3 cases of comparison and favorable result of comparison for 3 comparisons |
| 4 | The situation did not change in any of the comparisons |
| 5 | Positive result of comparing the indicator in 1 of 3 comparison cases and "without changes" for the other 2 comparisons, or Positive result of comparing the indicator in 2 of 3 cases of comparison and an unfavorable result of comparison for 3 comparisons |
| 6 | Positive result of comparing the indicator in 2 of 3 comparison cases and "without changes" for 3 comparisons |
| 7 | Positive result of comparing the indicator in 3 of 3 comparison cases |

Source: own elaboration.

Each of the financial indicators was assessed individually and could be rated from 1 to 7. The lowest score (1) was given to the indicators in which the impact on the bank's financial condition deteriorated in each of the three areas of comparison, while the highest (7), was given when there was an improvement in three areas. Grade 4 was awarded when no changes were observed for the entire comparison area.

In the third stage, on the basis of the Likert scale assigned to the individual indicators, the arithmetic average was calculated for each area of financial condition. The resulting construct, called the ABI (ABI—indicator for assessing the impact of using the BI system on the company's financial condition), can be used to synthetically assess the impact of using the BI system on specific areas, and consequently, on the overall financial condition of the organization:

$$ABI = \frac{\sum CG}{N},$$ (1)

where:

ABI—assessment of the impact of using the BI system on the financial condition of an enterprise;
CG—component grade based on the Likert scale;
N—number of financial condition indictors analyzed.
The limitations of the proposed method include two aspects:

1. The comparative method takes into account the impact of the implementation of the Business Intelligence against the background of general changes in the economic situation and changes in the financial condition of the closest competition (group of banks), however, the financial condition of the bank is also affected by many other factors that were not included in the method
2. The method does not take into account all cases of implementing the Business Intelligence system in banks, but focuses on specific banks (however, it includes 60% of the assets of commercial banks in Poland). It should be remembered that in other cases the impact of using BI on financial condition might have a different direction and strength.

Despite the above restrictions, the method has undoubted advantages—simplicity of application, universality and quantitative approach. Simplicity, because it requires uncomplicated calculations and comparisons. Universality, because the method can be used in any industry and not only in relation to the implementation of the BI system, but also of other potential determinants of financial conditions. Quantitative approach, because not only the chosen financial indicators, the component grades as well as the synthetic ABI indicator were calculated according to the described mathematical formulas, but also the specified materiality thresholds were applied. As a result, the synthetic indicator (ABI) was not unbiased and can clearly show when the implementation of Business Intelligence brings a positive or negative impact on a specific area of financial condition assessment. Owing to that method, the impact of implementing Business Intelligence on the financial condition can be easily and quickly calculated. What is more, by applying this method, managers can identify which areas of the financial condition may be affected by the Business Intelligence system. Having participated in many implementations of the BI system, the author has never come across any member of managerial staff of an organization testing such interaction. In the author's opinion, such a study could bring interesting conclusions about the development of the BI system in the future. It is also worth emphasizing that this method may also be applied in the assessment of any other IT system (apart from BI system) that may seem to possess an impact on the financial condition of an organization.

## 3. Results

The results of the conducted study were placed in separate sections, which correspond to the three perspectives of comparison used in the presented method. Detailed results are demonstrated in the tables and charts. However, the synthetic ABI indicator is described in the final section of this part of the article.

*3.1. The First Perspective of Comparison—to the Bank's Own Indicators before and after the Implementation of the Business Intelligence System*

In the first stage, the financial indicators were analyzed before the implementation of the BI system in relation to the period after the implementation of this system. The results are presented in Table 4.

**Table 4.** The first perspective of comparison of the bank's own indicators before and after the implementation of the Business Intelligence system.

| Area | Indicator | Banks | | | | | |
| --- | --- | --- | --- | --- | --- | --- | --- |
| | | **A** | **B** | **C** | **D** | **E** | **F** |
| Capital adequacy | Solvency ratio | P | P | N | N | P | P |
| Quality of assets and liabilities | Non-profit asset financing ratio | P | P | P | P | P | P |
| | Equity ratio | P | P | N | P | WC | P |
| | The share of loans in assets | P | P | P | P | WC | P |
| Liquidity | Coverage of receivables from customers | WC | P | P | P | P | N |
| | Cash payables ratio | N | P | N | WC | N | P |
| Productivity | Operating cost to assets ratio | P | P | P | P | P | WC |
| | Operating expense ratio to banking income | P | P | P | P | WC | WC |
| | Asset utilization rate | P | P | P | P | N | P |
| Profitability | Operating profit margin | P | P | P | P | N | P |
| | ROA | P | WC | P | WC | N | P |
| | ROE | P | P | P | P | N | P |
| Debt | Ratio of liabilities in equity | P | P | N | P | WC | P |
| | Customer payables ratio | P | WC | WC | WC | N | P |

Source: own elaboration; note: P—positive, WC—without changes, N—negative

As a result of the comparisons, a positive situation was observed among the vast majority of indicators (68 out of 84) after the implementation of the Business Intelligence system compared to the period before its implementation.

Considering the summary results from six banks, only one of the liquidity ratios, i.e., the cash payables ratio, indicated a negative situation after the implementation of the BI system (three negative observations, one without changes, two positive).

The highest number of positive observations of finance indicators in the period after the implementation of the BI system was observed in the areas of productivity, the quality of assets and liabilities and profitability. The total number of positive observations is presented in Figure 1.

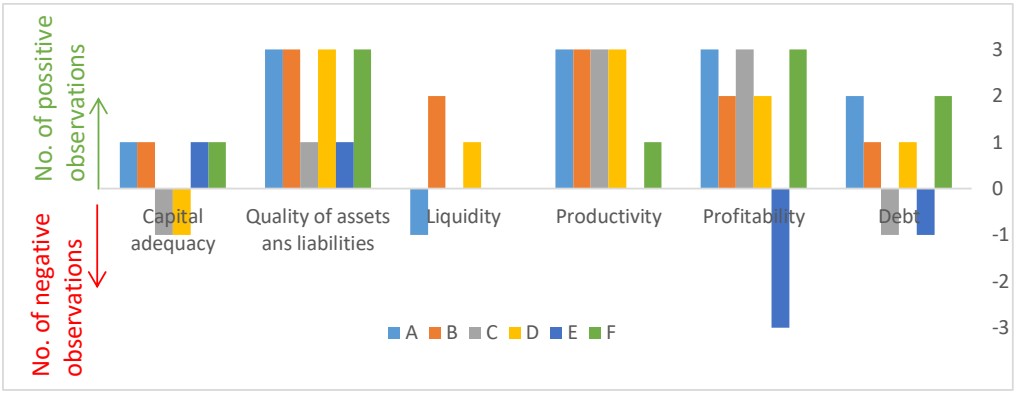

**Figure 1.** The number of positive and negative observations aggregated into groups of indicators—a comparison of bank indicators before and after the implementation of the Business Intelligence system.

The aggregated observation results for individual banks clearly indicate that the vast majority of the favorable situation was observed after the implementation of the BI system in relation to the period before its implementation. The lack of a bar for a particular bank on the chart means that the overall results do not indicate an improvement or deterioration of the situation.

*3.2. Second Comparison Perspective–Compared to the Results of a Group of Banks in the Period after the Implementation of the Business Intelligence System*

In the second stage, the ratios of six banks were examined in the period after the implementation of the Business Intelligence system in relation to the group of banks, defined as the closest competition of the research objects. The result is given in Table 5 below.

**Table 5.** The second perspective of comparison to the results of a group of banks in the period after the implementation of the Business Intelligence system.

| Area | Indicator | Banks | | | | | |
|---|---|---|---|---|---|---|---|
| | | **A** | **B** | **C** | **D** | **E** | **F** |
| Capital adequacy | Solvency ratio | P | WC | P | P | N | N |
| Quality of assets and liabilities | Non-profit asset financing ratio | P | P | P | P | WC | N |
| | Equity ratio | N | P | N | P | WC | N |
| | The share of loans in assets | N | N | N | WC | P | P |
| Liquidity | Coverage of receivables from customers | P | N | N | N | P | P |
| | Cash payables ratio | N | P | N | P | N | WC |
| Productivity | Operating cost to assets ratio | P | N | P | WC | WC | P |
| | Operating expense ratio to banking income | P | P | N | P | P | P |
| | Asset utilization rate | P | P | N | P | P | N |
| Profitability | Operating profit margin | P | P | N | P | P | N |
| | ROA | P | P | N | P | P | N |
| | ROE | P | P | N | P | P | N |
| Debt | Ratio of liabilities in equity | N | P | N | P | P | N |
| | Customer payables ratio | N | P | P | P | WC | N |

Source: own elaboration.

The most positive comparisons after the implementation period of the Business Intelligence system were observed among the banks A, B, D and E. In turn, the comparisons of indicators for banks C and F indicate that in most cases a negative situation was observed. Two areas of the indicators were distinguished by the overwhelming number of positive observations, including: the group of productivity and the profitability indicators.

The aggregated results of the comparison in relation to the individual areas of the indicators are presented in Figure 2. They show that a definite positive situation was observed in the levels of indicators in the area of productivity and profitability. A slightly less strong, but also favorable situation, was observed in the area of capital adequacy, quality of assets and liabilities as well as debt.

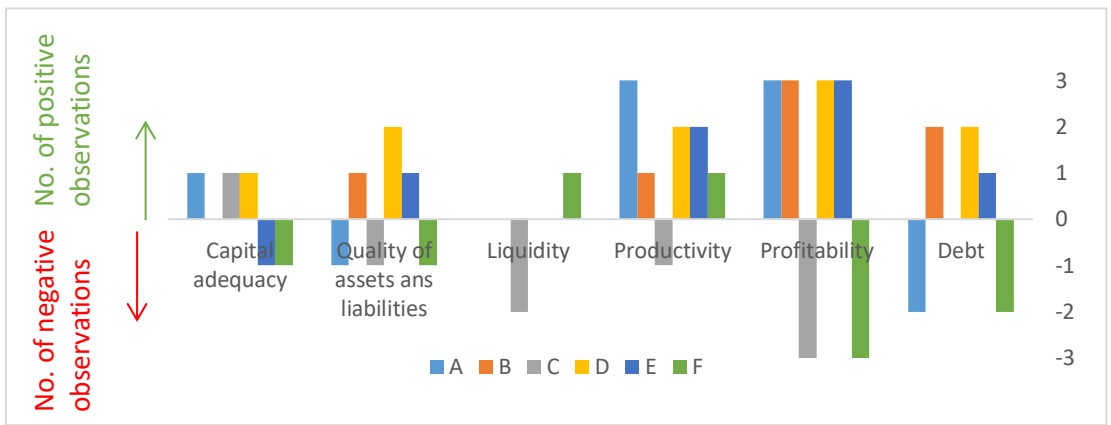

**Figure 2.** The number of positive and negative observations aggregated to the indicator areas—a comparison with a group of banks after the implementation of the Business Intelligence system.

### 3.3. The Third Perspective of Comparison—In Relation to the GDP Dynamics after the Implementation of the Business Intelligence System

The last, third perspective of comparison in relation to the GDP dynamics in the period after the implementation of the Business Intelligence system is presented in Table 6 below.

**Table 6.** The third perspective of comparison to GDP dynamics after the implementation of the Business Intelligence system.

| Area | Indicator | Banks | | | | | |
|---|---|---|---|---|---|---|---|
| | | **A** | **B** | **C** | **D** | **E** | **F** |
| Capital adequacy | Solvency ratio | WC | WC | N | N | WC | WC |
| Quality of assets and liabilities | Non-profit asset financing ratio | P | P | WC | P | WC | N |
| | Equity ratio | WC | WC | WC | N | N | P |
| | The share of loans in assets | N | WC | P | WC | N | N |
| Liquidity | Coverage of receivables from customers | N | WC | P | WC | WC | N |
| | Cash payables ratio | P | P | N | N | WC | P |
| Productivity | Operating cost to assets ratio | P | P | P | P | P | P |
| | Operating expense ratio to banking income | P | P | P | P | N | WC |
| | Asset utilization rate | WC | WC | WC | WC | N | P |
| Profitability | Operating profit margin | P | WC | P | WC | N | P |
| | ROA | WC | WC | WC | N | N | N |
| | ROE | N | N | N | N | N | N |
| Debt | Ratio of liabilities in equity | P | WC | N | WC | P | P |
| | Customer payables ratio | WC | WC | N | N | N | P |

Source: own elaboration.

A comparison with the GDP dynamics in the period after the implementation of the Business Intelligence system indicates an ambiguity of the results. For some banks, i.e., banks A, C and F, most comparisons show an improvement for banks D and F and most indicate deterioration, while for bank B, the number of positive and negative comparisons is the same.

Considering the total number of comparisons for the six banks, which were presented in Figure 3, a positive situation was observed in the area of indicators related to productivity. For five out of six banks, they indicated a favorable situation after the implementation of the Business Intelligence system. The opposite situation was observed in the area of profitability and capital adequacy, while in other areas neither improvement nor deterioration could be clearly stated.

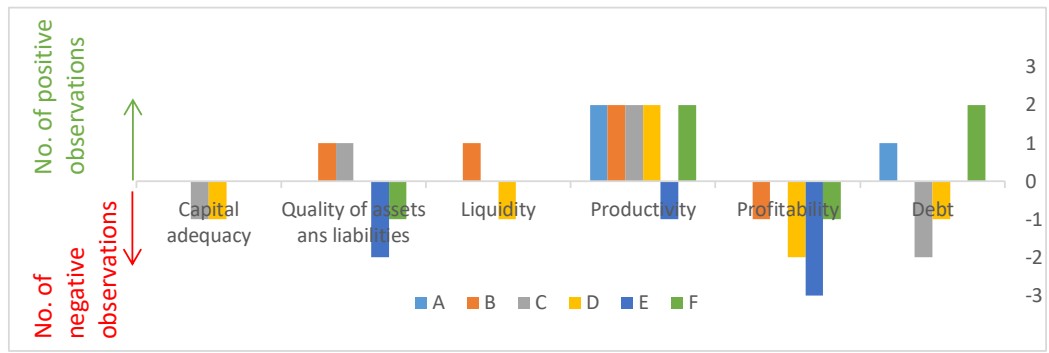

**Figure 3.** The number of positive and negative observations aggregated to the indicator areas—a comparison in relation to the GDP dynamics after the implementation of the Business Intelligence system.

### 3.4. Synthetic ABI Indicator–Assessment of the BI System Impact on Six Commercial Banks

Based on the detailed test results and the individual component grades (positive = 1, without changes = 0, negative = −1), each indicator was awarded grades from 1 to 7 according to the Likert scale discussed earlier. Then, the ABI index (impact assessment of the use of the Business Intelligence

system) was calculated for each of the six banks and in each of the analyzed financial indicator areas. The results of the calculations are presented in Table 7 below.

**Table 7.** Indicator for assessing the impact of using the BI system on the company's financial condition indicator (ABI)—assessment of the impact of using the BI system on the financial condition of an enterprise.

| Area | Indicator | A | B | C | D | E | F |
|---|---|---|---|---|---|---|---|
| Capital adequacy | Solvency ratio | 6 | 5 | 3 | 3 | 4 | 4 |
| Quality of assets and liabilities | | 4.7 | 5.7 | 4.3 | 5.7 | 4.0 | 4.3 |
| | Non-profit asset financing ratio | 7 | 7 | 6 | 7 | 5 | 3 |
| | Equity ratio | 4 | 6 | 2 | 5 | 3 | 5 |
| | The share of loans in assets | 3 | 4 | 5 | 5 | 4 | 5 |
| Liquidity | | 3.5 | 5.5 | 3.0 | 4.0 | 4.0 | 4.5 |
| | Coverage of receivables from customers | 4 | 4 | 5 | 4 | 6 | 3 |
| | Cash payables ratio | 3 | 7 | 1 | 4 | 2 | 6 |
| Productivity | | 6.7 | 6.0 | 5.3 | 6.3 | 4.3 | 5.3 |
| | Operating cost to assets ratio | 7 | 5 | 7 | 6 | 6 | 6 |
| | Operating expense ratio to banking income | 7 | 7 | 5 | 7 | 4 | 5 |
| | Asset utilization rate | 6 | 6 | 4 | 6 | 3 | 5 |
| Profitability | | 6.0 | 5.3 | 4.0 | 5.0 | 3.0 | 3.7 |
| | Operating profit margin | 7 | 6 | 5 | 6 | 3 | 5 |
| | ROA | 6 | 5 | 4 | 4 | 3 | 3 |
| | ROE | 5 | 5 | 3 | 5 | 3 | 3 |
| Debt | | 4.5 | 5.5 | 2.5 | 5.0 | 4.0 | 5.0 |
| | Ratio of liabilities in equity | 5 | 6 | 1 | 6 | 6 | 5 |
| | Customer payables ratio | 4 | 5 | 4 | 4 | 2 | 5 |

Source: own elaboration.

An indicator value above 4 (marked in green) indicates that a favorable (positive) situation was observed after the implementation of the Business Intelligence system, while the value below 4 is the opposite.

Among the analyzed areas, one—the area of productivity indicators—stands out. The ABI indicator shows an improvement in the situation after the implementation of the Business Intelligence system for each bank. An almost equally favorable situation was observed in the area of quality and liabilities (only bank E has an ABI value equal to 4). An ambiguous situation was observed in the area of liquidity and capital adequacy, where two banks indicated an improvement, two no change, and two indicated a deteriorated situation after the implementation of the Business Intelligence system.

In the last step, the ABI index values from all the banks in each area were averaged. This summary is presented in Figure 4 below.

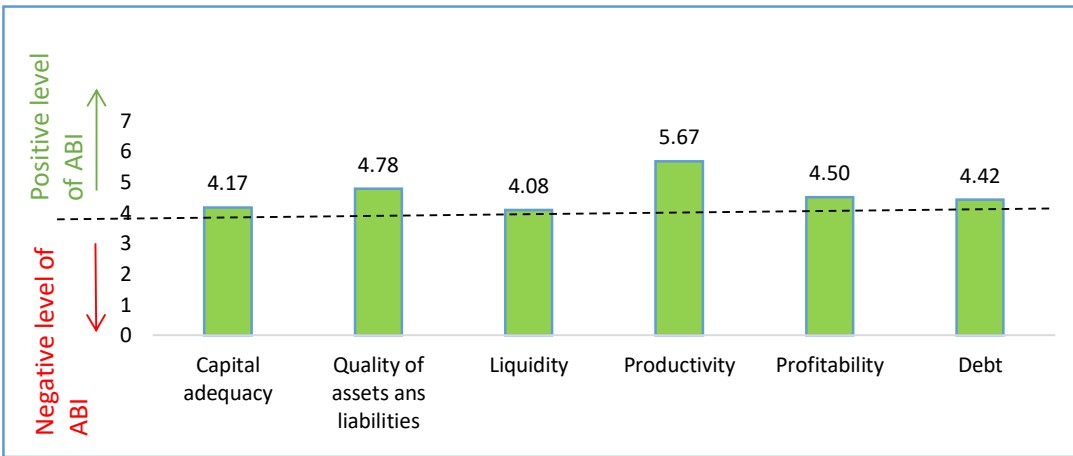

**Figure 4.** Synthetic value of the ABI comparison in the six analyzed banks.

To sum up, it is worth noting that the synthetic value of the ABI index for all six banks in all areas is above 4.0. This means that an improvement in the financial indicators was observed in all areas of the financial condition after the implementation of the Business Intelligence system. By far the most favorable situation was observed in the area of productivity, where the average ABI index for all banks was 5.67. In the area of asset and liability quality (4.78), profitability (4.5), as well as the area of debt (4.42), significantly high ABI values were also observed. This means that the implementation of the BI system can have a positive impact in these areas of a bank's financial condition. In turn, in the areas of capital adequacy (4.17) and liquidity (4.08), the value of the ABI indicator demonstrates a slight improvement in the indicators compared to the period before the implementation of the BI system.

## 4. Discussion

As a technology of top priority resulting from its ability to provide in-depth knowledge for decision-making processes, Business Intelligence systems have managed to attract the major attention of policy makers and industry experts [34].

The empirical research published to date regarding the impact of Business Intelligence systems has mainly concerned enterprises, not banks, and has been conducted outside of Poland. Most studies on the impact of BI on the condition of an organization indicate its positive aspect. It can therefore be assumed that this relationship should be similar in banks. The analysis of the literature also shows that the application of the system grows every year. BI systems have been increasingly used in new industries while the industries already using the systems have found their new applications. Previous studies of the implementation of Business Intelligence systems indicate that the results of using the BI system are hard to measure and are achieved mainly in the long term [6,56]. The results of previous research prompted the author to investigate the difference in the financial condition indicators of banks that decided to implement the Business Intelligence system.

To date, the methods of assessing the impact of BI presented in literature are mostly qualitative or quantitative. However, no simple uniform method of assessing the impact of BI on the condition of an organization has been identified as a result of known research. This study has attempted to enrich the recent literature by describing a novel comparative method of analyzing the changes in the financial condition of banks following the implementation of the BI system. It is also of importance that studies on the impact of BI on enterprises in Poland have mainly included case studies [45–47] and there have been a few publications concerning the long-term studies assessing the impact of BI systems on organizations [44].

The identified results of the study show positive long-term relation between implementing Business Intelligence systems and the financial condition of commercial banks in Poland in all the examined areas. The results of this study may suggest a direction of development in the banking sector in the area of Business Intelligence systems. The demonstrated positive relationship between the use of the Business Intelligence systems and financial condition may be an indication for bank managers that investments in the Business Intelligence system improve the financial condition. The research results provide information that productivity, profitability and debt were subsequently the three main areas most beneficially influenced by the implementation of the Business Intelligence system. This is a valuable suggestion for managers responsible for the development of Business Intelligence systems, regarding the areas where, in banks, the Business Intelligence system would be worth implementing or developing in the first place. In turn, for those banks that do not yet have this type of system, the above research results should be a premise for considering an investment in the Business Intelligence system. From the experience of the author who participated in many implementations of the BI system, attempts to examine the impact of the BI system by organization managers have only rarely taken place. In the author's opinion, such a study could bring an impulse to the development of the BI system in a given organization. It is also worth emphasizing that the proposed evaluation methodology might also be applied in the assessment of any system implemented to expand analytical competences (not only

Business Intelligence), where there is a reasonable premise that such a system has an impact on the financial condition of an organization.

Since the impact of the BI system has been observed at the level of financial condition, it is worth establishing a causal relationship and explaining the reason for the observed research results. The complexity of banking operations and the guidelines of the supervising institutions stating that a bank should have a defined structure, as well as implementing and regularly evaluating procedures, force banks to operate in a process model. Three main groups of processes can be distinguished in banks: management processes, operational processes and supporting processes. Then, the groups of processes can be divided into several metaprocesses such as strategic management, risk management and capital adequacy, sales management and product administration, marketing and sales management, providing accounting services, IT management, human resources management, financial and accounting services, tax services, administration and logistics management, crime prevention and other support processes. Subsequently, many processes and subprocesses can be assigned to each metaprocess [59]. In day-to-day operations, many of these processes are carried out by IT systems. These systems collect data on the implementation of individual banking operations and such activities as, for example, cash withdrawal, making a transfer or requesting a loan.

In the author's assessment, the processing of the data described above can be used to improve, support or further the automatic implementation of the listed processes. These activities can be carried out globally in one system, which is a Business Intelligence class system, while providing a number of benefits resulting from new analytical capabilities. It is worth recalling here the basic value from the use of the Business Intelligence system. It indicates that the implementation of the BI system causes a "transformation" of an organization, triggers an impulse for changes, as well as gives completely new possibilities of information processing [44,60].

Therefore, a causal relationship between the impact of providing information by Business Intelligence class systems on individual banking processes and subprocesses is justified, and in turn it contributes to the improvement of the bank's financial condition indicators.

However, the obtained test results are not without restrictions. The impact of implementing the Business Intelligence system against the background of general changes in the economic situation and changes in the financial condition of the nearest competition have been taken into account. Nonetheless, a bank's financial condition is also affected by many other factors that have not been included in the method. The financial condition of the organization is affected by other factors that can be divided into external, such as government policy, economic situation, cost of capital, interest rates and internal ones including earned revenues and incurred costs as well as the quality of management, including systems supporting information management processes. Another limitation of the comparative method is the fact that it does not cover all the cases of implementing the BI system in banks, but focuses on specific banks (despite the fact that the six chosen banks represent 60% of the assets of commercial banks in Poland). It should be remembered that in other cases the impact of using BI on financial condition might have a different direction and strength.

Despite the above restrictions, the proposed comparative method has unquestionable advantages—simplicity of application, universality and quantitative approach. Simplicity, because it requires uncomplicated calculations and comparisons, universality, because the method can be applied to other industries and quantitative approaches, because all the components as well as synthetic ABI indicator were calculated according to the described mathematical formulas. As a result of its application, the managers of organizations can quickly find out about the impact of the implementation of the Business Intelligence system on the financial condition.

## 5. Conclusions

This study investigated the relationship between the impact of using the Business Intelligence system in an organization and its financial condition. The data from the commercial banks listed at the Warsaw Stock Exchange in 2020, which have implemented the Business Intelligence system since

2001, provided the background for this study. The six banks chosen out of the thirteen listed in 2020, represent 60% of assets of commercial banks in Poland. The comparative method designed by the author was based on three parallel comparisons of each of the indicators of the financial condition: before and after the implementation of the BI system (comparison over time), with average indicators for the group of banks (industry comparison), with reference to the changes of the general economic situation (GDP). All the calculations were based on the specified set of the 14 indicators of the bank's financial condition that were divided into six categories: liquidity, quality of assets and liabilities, debt, productivity, profitability and capital adequacy.

The obtained results confirm the research hypothesis and indicate a general improvement in the overall financial condition of banks as a result of using the BI system. The positive impact of using the BI system on financial condition indicators was identified in all the examined areas of the financial condition. However, the most noticeable impact relates to the areas of productivity, quality of assets and liabilities, profitability and debt. The fact that the surveyed banks hold a total of 60% of the assets of all commercial banks, as mentioned above, justifies the generalization of results for the commercial banks sector in Poland. The hypothesis that the Business Intelligence system provides sustainability to the commercial banks by influencing the financial condition has been confirmed.

The positive results of the study concerning the banking industry also suggest that to consider a research on the impact of using BI systems in other industries would be worthwhile. This is going to be the subject of the author's further scientific activity.

**Funding:** This research received no external funding. The APC was funded by WSB University in Gdansk.

**Conflicts of Interest:** The author declares no conflict of interest.

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
