# Peer review of "Sustainability of Commercial Banks Supported by Business Intelligence System"

_sustainability, doi:10.3390/su12114754_

Round 1

Reviewer 1 Report

Report on “Sustainability of Commercial Banks Supported by 3 Business Intelligence System”

The article is focused on establishing whether Business Intelligence systems provide sustainability to commercial banks by influencing their financial condition. As part of the search for a solution to the research problem,

The authors formulates a hypothesis which assumes that the use of the Business Intelligence management system improves the financial condition of commercial banks. To assess the impact, they compare financial condition indicators in three aspects: before and after the implementation of the BI system (comparison over time), with average indicators of a group of banks (comparison to the industry), with reference to changes in the overall economic situation. They computer a synthetic indicator of the impact of using Business Intelligence (ABI) and conduct their study in the six out of thirteen largest commercial banks, listed on the Warsaw Stock Exchange in 2020, which have implemented the Business Intelligence system since 2001. They find a positive impact of using the BI system on selected areas of the financial condition of commercial banks and the impact is related to productivity, quality of assets and liabilities, profitability and debt.

I find their findings interesting. I have some comments to the authors to improve their paper:

  • The authors should discuss more on the literature review, discuss all the important works related to their study.
  • The authors should point out the contributions of their study to the literature that are different from the literature.
  • The authors should discuss some related theories that support their study and build up a theory to support their study.
  • The authors did define the variables used in their paper but they have not stated all the related econometric models to be used. They should discuss the econometric models that they are using in their paper.
  • They should also conduct some proper tests to their results, not only produce the estimates.
  • The authors should discuss how investors, bankers, and policymakers can use their findings to make better decisions.

Reviewer 2 Report

Dear author,

your research faces an interesting topic these days. Artificial intelligence systems implementation is a practice that is driving to better financial results and banks are no exception. However, I wonder how you verify de causality relation between such factors. are more suitable banks who implement an artificial intelligence system or viceversa? 

Although methods sound correct there is not the only one to test this relation. Why did you not use others? Please, justify.

In addition, this study is performed over Polish bank's data. how the cultural environment could be affecting results? Please, point out some ideas about this concern.

Regards

Reviewer 3 Report

Dear author,

Thank you for giving me the opportunity to review your paper, "Sustainability of Commercial Banks Supported by Business Intelligence System".

The paper is based on the impact of Business Intelligence (BI) on commercial banks' sustainability. The main hypothesis is designed to test whether BI improved financial conditions of the tested subjects, by comparing selected financial ratios of listed banks on the Warsaw Stock Exchange, before and after the BI implementation, and proposing a synthetic indicator to measure this impact.

The paper offers the author's perspective on how this research dilemma may be solved, focusing on three steps process. 

The topic is of significant interest to the research community and experts in the industry and I salute the author's efforts and contribution.

Please find some specific recommendations that may improve the quality of your paper:

  1. The literature review may be enhanced with other relevant, recent articles dedicated to the impact of BI in different industries and its way to reshape business framework & models; other similar research efforts in the banking industry worldwide may provide a plus;
  2. The research methodology and procedures may be improved by focusing on more details of how the assessment of "positive (P)", "without change (WC)" and "negative N" was performed and its materiality threshold;
  3. The synthetic indicator seems to be based more on a judgemental approach, based upon the author experience; additionally, an unbiased, quantitative validation would bring more value-added to the scientific endeavor;
  4. In the results section, additional comments on how this synthetic indicator would serve the overall banking industry may be helpful;
  5. The limitations of the research may be presented more extensively in the conclusions.

Overall I consider the topic relevant for publication. Nevertheless, improvements are needed, in order to strengthen the research quality.

Round 2

Reviewer 1 Report

I find the paper is of good quality with interesting findings. I am satisfied with his revision, and thus, I suggest accepting the paper.

Author Response

Thank for all comments and suggestions.